# [RE] CNN-generated images are surprisingly easy to spot... for now

## Reproducibility Summary

*This work evaluates the reproducibility of the paper "CNN-generated images are surprisingly easy to spot... for now" by Wang et al. published at CVPR 2020. The paper addresses the challenge of detecting CNN-generated imagery, which has reached the potential to even fool humans. The authors propose two methods which help an image classifier to generalize from being trained on one specific CNN to detecting imagery produced by unseen architectures, training methods, or data sets.*

**Scope of Reproducibility**

The paper proposes two methods to help a classifier generalize: (i) utilizing different kinds of data augmentations and (ii) using a diverse data set. This report focuses on assessing if these techniques indeed help the generalization process. Furthermore, we perform additional experiments to study the limitations of the proposed techniques.

**Methodology**

We decided to implement the methods from scratch to highlight possible hurdles for practitioners who want to adopt the results. For our experiments, we utilized the training and test data provided by the authors, as well as, creating our own data set to analyze how the proposed techniques perform on different data sets. Note that overall we estimate the entire experiments to take around 590 GPU hours.

**Results**

In general, we were able to replicate the overall results reported in the paper. However, we also found significant differences in our experiments. After further investigations, we discovered that we implemented two data augmentations slightly different. Additionally, we identified three test data sets, which exhibit fluctuating results across multiple runs with the same setup. However, the overall trends of our experiments matched the original publication, i.e., data diversity and augmentations help generalization. We also performed several additional experiments on other data sets, highlighting limitations and clarifying the methods.

**What was Easy**

The paper is very detailed and we could faithfully replicate the experiments (bar the aforementioned exception). We only utilized the authors' code sparingly, but the repository is very well documented and provides pre-trained models.

**What was Difficult**

The naming of one of the augmentations caused confusion on our part. Note that the description in the paper is correct, however, the naming suggest the method works different than actually implemented. This resulted in us implementing a variation of the proposed technique. Surprisingly, our method actually improved on the original results.

**Communication with Original Authors**

Most of our question could be resolved by comparing our implementation against the authors' code. We contacted the first author regarding our different results, he was very responsive and answered all of our questions.

## 1 Introduction

High-quality implementations of state-of-the-art image synthesis methods are available to the general public [Karras et al., 2020]. This technology is astonishing and concerning at the same time. One the one hand, such algorithms can be used for practical use cases like image-to-image translation [Isola et al., 2017, Liu et al., 2017, Zhu et al., 2017] or image super-resolution [Lai et al., 2017, Ledig et al., 2017, Lim et al., 2017]. On the other hand, they can generate images which even trick human observers [Simonite, 2019]. Recently, fake news and tampered media have become practical problems which even have the potential to influence democratic processes [Thompson and Lapowsky, 2017, Hao, 2019].

Recognizing these problems, Wang et al. [2020] propose a "universal" detector for CNN-generated images. While creating a detector for know architectures has been done before [Zhang et al., 2019, Wang et al., 2019a], the authors of this paper ask whether it is conceivable to train a classifier which generalizes to new, unseen architectures. The authors demonstrated that with a diverse data set and with carefully chosen data augmentation methods, a standard image classifier can achieve this goal. More specifically, they train a ResNet50 [He et al., 2016] classifier on a data set consisting of images generated by 20 different instances of ProGAN [Karras et al., 2018]. Additionally, they augment the training process by utilizing aggressive image filtering techniques (i.e., Gaussian Blur and JPEG compression). Combining both techniques allows them to create an image classifier which even generalized to StyleGAN2 [Karras et al., 2020], which was published concurrently.

**Scope of reproducibility:** We opted to reimplement the methods from scratch, aiming at highlighting difficulties for practitioners who want to adapt these methods. While the authors released a very well documented code repository, we initially abstained from inspecting the source code to not prime ourselves. In the later stages of our analysis, we used their repository to uncover some small implementation differences, which lead to our implementation actually improving on the original results.

Using our implementation, we aim at verifying the two main claims of the paper:

1. The proposed data augmentations (usually) improve generalization.
2. More diverse data sets help the classifier generalize to unseen architectures.

We recreate parts of the original experiments which directly investigate these two claims (see Section 4.2, Section 4.3, Table 2, and the Figures 2 and 3 in the original paper [Wang et al., 2020]). Departing from their work, we perform several additional experiments: First, we further investigate the first claim, training multiple additional data set combinations to clarify differences between our results and the original paper. Second, we generate a new training set from a different CNN generator (StyleGAN2 [Karras et al., 2020]) and evaluate if the results transfer. Finally, we investigate if the method is exclusive to the specific image classifier used in the paper. To this end, we train two other classifiers and compared them to the original results: A different image classifier architecture [Simonyan and Zisserman, 2015] and a classifier which leverages spectral analysis [Frank et al., 2020].

## 2 Methodology

In this section, we provide an overview of our methodology. We start by presenting the different data sets used in our experiments. Then, we describe the overall experimental setup, including the models we studied, data augmentations performed, details on the training procedure, our evaluation criteria, and computational requirements. All of our reproducibility efforts are based on the arXiv version of the paper (v2 – submitted 04. April 2020) and the corresponding GitHub repository[1] (commit f692c13 – 26. October 2020). We also make our implementation, our pre-trained models and our data set publicly available [2].

### 2.1 Data Sets

For training our networks, we used two different data sets: The first one is provided by the original publication. The authors used 20 ProGAN [Karras et al., 2018] models, each trained on a different LSUN category [Yu et al., 2015], to sample 18K generated images per model. They also collected 18K training images for each of the corresponding classes, resulting in a total data set of $(18.000 + 18.000) \cdot 20 = 720.000$ images.

Additionally, we created a new data set to study the proposed techniques on a different base generator. We downloaded the pre-trained models for StyleGAN2 [Karras et al., 2020]. Using these models, we generated a training set of 36K

---

[1]available at `https://peterwang512.github.io/CNNDetection/`
[2]available at `https://github.com/mlreprochallenge/CNNEasyToSpot`

train (18K fake and 18K real) images for the LSUN *cat and horse* data sets, following the pre-/post-processing steps outlined in the original publication. Since these categories are also contained in the data set by Wang et al., this allows for a one-to-one comparison.

We exclusively used the test data set provided by the authors. The data set consists of images from 11 different synthesis models:

- GANs: The data set contains samples from six different Generative Adversarial Networks (GANs): Pro-GAN [Karras et al., 2018], StyleGAN [Karras et al., 2019], BigGAN Brock et al. [2019], GauGAN [Park et al., 2019], CycleGAN [Zhu et al., 2017], and StarGAN [Choi et al., 2018].

- Perceptual loss: The authors also include samples from generative models which are directly trained to optimize a perceptual loss: Cascaded Refinement Networks (CRN) [Chen and Koltun, 2017] and Implicit Maximum Likelihood Estimation (IMLE) [Li et al., 2019].

- Image manipulation models: It includes samples from Seeing In The Dark (SITD) [Chen et al., 2018], which approximates long-exposure photography, and the Second-Order Attention Network (SAN) [Dai et al., 2019], which generates images at a higher resolution.

- Deep fakes: Finally, the authors include samples from the FaceForensics++ [Rössler et al., 2019] data set.

## 2.2 Experiment Setup

In the following, we provide an overview of the different experiments performed in the paper. We start by discussing the different models used, present the data augmentations proposed by Wang et al., and, proceed by presenting the training schedule used in all experiments. Finally, we conclude by discussing how the evaluation and with an overview of the computational requirements needed for reproducing this report.

**Models used:** Following the original paper, we use a ResNet50 [He et al., 2016] model as the image classifier for the majority of our experiments. The classifier is pre-trained on ImageNet [Russakovsky et al., 2015]. We use random crops (to $224 \times 224$ pixels) and horizontal flipping during training of all of our classifiers. Additionally, we scale the image to the range $[0, 1]$, remove the mean, and scale them to unit variance.

For the experiments presented in Section 3.3.2, we also train a VGG-11 [Simonyan and Zisserman, 2015] model and a classifier which utilizes spectral information [Frank et al., 2020]. The VGG model is also pre-trained on ImageNet and uses the same preprocessing as our ResNet models. The spectral classifier also uses random crops and horizontal flipping as outlined above. Afterwards, we follow Frank et al. and use Discrete Cosine Transfrom-II [Ahmed et al., 1974] (DCT-II) to transform the images from the spatial to the spectral domain. Afterwards, we use MinMax-scaling to scale the images to the range $[0, 1]$ and train a ResNet50 model from scratch on the transformed data.

**Data augmentations:** Wang et al. proposed several data augmentations which help their classifier generalize to unseen architectures:

- *Gaussian blur (Blur):* Before cropping, with 50% probability, the images are blurred with $\sigma \sim \text{Uniform}\{0, 3\}$

- *JPEG-compressiong (JPEG):* with 50% probability images are JPEG-ed by two popular image libraries, OpenCV and PIL, with quality $\sim \text{Uniform}\{30, \ldots, 100\}$.

- *Blur + JPEG (Blur & JPEG (0.5)):* images are possibly blurred and JPEG-ed, each with 50% probability.

- *Blur + JPEG (Blur & JPEG (0.1)):* similar to the previous, but the probability is dropped to 10%.

All augmentations are applied before cropping and flipping (to the entire training set). Additionally, we also evaluate no augmentations (*No Aug.*) as a baseline. Note that the JPEG standard only specifies quality setting in the range $[30, 95]$. However, the authors sample in the range $[30, 100]$. We follow their sampling, but clip all values to the allowed range.

**Training details:** At the start of the training, we randomly sample $10\%$ of the training set for validation. We train all model using Adam [Kingma and Ba, 2015] with $\beta_1 = 0.9$, $\beta_2 = 0.999$, a batch size of 64, and an initial learning rate of $10^{-4}$. We drop the learning rate by $10\times$ when the validation accuracy stagnates for 5 epochs. When the learning rate reaches $10^{-7}$, we terminate training and select the best classifier based on the validation accuracy.

We abstained from validating additional combinations of (Adam-) hyperparameters, learning rate, and batch size. The selection of data augmentation and data set variety is already computational challenging and can be viewed as a hyperparameter in its own right. However, we note that this might be an interesting direction for future experiments.

Table 1: **Average Precision of training a ResNet50 classifier on the entire data set with different augmentations.** We report the average precision for both our results (Reproduced) and the the original publication (Wang et al.). Additionally, we report the results of a (Corrected) implementation of the Blur + JPEG augmentation. Chance is 50%, the best possible results is 100%, and we highlight the ProGAN results in gray since the classifier is trained on similar data. We train on the entire data set (20 classes).

| Name | Result | ProGAN | StyleGAN | BigGAN | CycleGAN | StarGAN | GauGAN | CRN | IMLE | SITD | SAN | DeepFake | **mAP** |
|---|---|---|---|---|---|---|---|---|---|---|---|---|---|
| No Aug. | Wang et al. | 100.0 | 96.3 | 72.2 | 84.0 | 100.0 | 67.0 | 93.5 | 90.3 | 96.2 | 93.6 | 98.2 | 90.1 |
| | Reproduced | 100.0 | 96.8 | 73.5 | 81.9 | 100.0 | 68.2 | 95.1 | 88.8 | 97.1 | **87.2** | 98.4 | 89.7 |
| Blur | Wang et al. | 100.0 | 99.0 | 82.5 | 90.1 | 100.0 | 74.7 | 66.6 | 66.7 | 99.6 | 53.7 | 95.1 | 84.4 |
| | Reproduced | 100.0 | **94.0** | *71.7* | 81.9 | 100.0 | 71.0 | **63.5** | 63.1 | **93.8** | *83.9* | 98.6 | 83.8 |
| JPEG | Wang et al. | 100.0 | 99.0 | 87.8 | 93.2 | 91.8 | 97.5 | 99.0 | 99.5 | 88.7 | 78.1 | 88.1 | 93.0 |
| | Reproduced | 100.0 | 99.4 | 89.4 | 93.3 | 94.5 | 96.4 | 99.1 | 99.3 | 91.1 | **69.7** | 92.5 | 93.2 |
| Blur + JPEG (0.5) | Wang et al. | 100.0 | 98.5 | 88.2 | 96.8 | 95.4 | 98.1 | 98.9 | 99.5 | 92.7 | 63.9 | 66.3 | 90.8 |
| | Reproduced | 100.0 | 99.4 | 88.8 | 94.1 | 96.7 | 96.2 | 98.5 | 99.1 | 92.9 | **72.3** | *93.7* | 93.8 |
| | Corrected | 100.0 | 99.1 | 88.8 | 94.8 | 94.5 | 97.5 | 99.3 | 99.4 | **84.8** | *74.1* | 73.0 | 91.4 |
| Blur + JPEG (0.1) | Wang et al. | 100.0 | 99.6 | 84.5 | 93.5 | 98.2 | 89.5 | 98.2 | 98.4 | 97.2 | 70.5 | 89.0 | 92.6 |
| | Reproduced | 100.0 | 99.5 | 85.7 | 93.3 | 97.9 | 93.6 | 99.2 | 99.4 | 95.5 | **77.7** | **95.0** | 94.2 |
| | Corrected | 100.0 | 99.6 | 86.5 | 94.7 | 97.1 | 92.5 | 99.1 | 99.4 | 95.3 | *82.2* | **96.9** | 94.8 |

We highlight differences (w.r.t the original publication) greater 5% in **bold** and greater 10% additionally in ***cursive***.

We use the same settings for training the VGG network. When training the DCT-ResNet we change the initial learning rate to $10^{-3}$ since we train the underlying model from scratch.

**Evaluation criteria:** We follow the authors and use Average Precision (AP) as our evaluation metric. It is a ranking-based metric which is not sensitive to the "base rate" of the fraction of fake images and thus commonly used throughout the literature [Zhou et al., 2018, Huh et al., 2018, Wang et al., 2019b]. Additionally, we compute the mean Average Precision (mAP) over all data sets, to compute an overall tendency. During testing, all images are center cropped without resizing to match the training dimension (without augmentations).

**Computational requirements:** All our experiments are run on two desktop machines. Each machine is running Ubuntu 18.04, with 64GB RAM, a AMD Ryzen 7 3700X 8-Core Processor, and two GeForce RTX 2080Ti. Training a single ResNet50 model on one GPU on the entire data set (20 classes) takes around 42 hours. Most of the experiments are conducted on smaller data set (2–8 classes), which significantly reduces the training time to roughly 6 hours. Overall, we estimate the entire experiments to take around 590 GPU hours, not including initial testing runs.

# 3 Results

Overall we were able to reproduce the trends of the original results and verify the two claims of the publication. In Section 3.1, we examine the first claim, studying in detail the different data augmentations proposed by the authors. In Section 3.2, we examine the second claim, evaluating if more diverse data help generalization. Finally, in Section 3.3, we depart from the original experiments and investigate further to gain additional insights into the proposed methods.

## 3.1 Claim: *Data augmentations improve generalization*

First, we investigate the claim that data augmentations improve generalization and reproduce the second half of Table 2 in the original publication. Specifically, we train five classifier on the entire training set, with each classifier trained on one data augmentation introduced in Section 2.2. The results are presented in Table 1.

**Results:** We can successfully recreate the general trend that data augmentations help generalization. However, for specific cases our results differ heavily from the reported results by Wang et al.. This is especially noticeable for the blur augmentation, where our results differ in multiple data sets (StylGAN, BigGAN, CycleGAN, CRN, SITD and SAN). The results for the SAN data set even differ by $30.2\%$. We also seem to slightly outperform the results by Wang et al..

We verified our implementation against the original published code and noticed two substantial differences: First, we implemented Gaussian filtering with the build-in methods provided by PyTorch [Paszke et al., 2019], while Wang et al. build their own method using SciPy [Virtanen et al., 2020]. Second, we apply blurring in conjunction with JPEG compression with a probability of 50%. In contrast, Wang et al. first apply blurring with a probability of 50% and *then* apply JPEG with a probability of 50%. Note that the paper describes this correctly.

Table 2: **Average Precision (AP) of training a ResNet50 classifier on different subsets of the training data.** We report the average precision over 11 different CNN generators for both, our results (Reproduced) and the results from the original publication (Wang et al.). Chance is 50% and the best possible results is 100%. Note that the classifier is trained on ProGAN and we thus display the results in gray. The mean Average Precision (mAP) is obtained by taking the mean of the individual APs. Note that we apply blurring and JPEG compression with 50% probability during training.

| Setting | Result | ProGAN | StyleGAN | BigGAN | CycleGAN | StarGAN | GauGAN | CRN | IMLE | SITD | SAN | DeepFake | **mAP** |
|---|---|---|---|---|---|---|---|---|---|---|---|---|---|
| 2-class | Wang et al. | 98.8 | 78.3 | 66.4 | 88.7 | 87.3 | 87.4 | 94.0 | 97.3 | 85.2 | 52.9 | 58.1 | 81.3 |
|  | Reproduced | 99.3 | 81.0 | **71.8** | 88.4 | 83.5 | 89.7 | 98.8 | 99.5 | 82.0 | *72.3* | 61.5 | 84.3 |
| 4-class | Wang et al. | 99.8 | 87.0 | 74.0 | 93.2 | 92.3 | 94.1 | 95.8 | 97.5 | 87.8 | 58.5 | 59.6 | 85.4 |
|  | Reproduced | 99.8 | 90.6 | **79.7** | 92.5 | 91.2 | 93.5 | 98.1 | 98.7 | **95.0** | *74.6* | *75.7* | 90.0 |
| 8-class | Wang et al. | 99.9 | 94.2 | 78.9 | 94.3 | 91.9 | 95.4 | 98.9 | 99.4 | 91.2 | 58.6 | 63.8 | 87.9 |
|  | Reproduced | 100.0 | 96.7 | 81.8 | 92.7 | 92.6 | 95.0 | 98.2 | 99.3 | **85.7** | *69.8* | *77.9* | 90.0 |
| 16-class | Wang et al. | 100.0 | 98.2 | 87.7 | 96.4 | 95.5 | 98.1 | 99.0 | 99.7 | 95.3 | 63.1 | 71.9 | 91.4 |
|  | Reproduced | 100.0 | 98.5 | 87.0 | 94.5 | 95.0 | 96.8 | 99.2 | 99.3 | *84.5* | *77.9* | 76.8 | 91.8 |
| 20-class | Wang et al. | 100.0 | 98.5 | 88.2 | 96.8 | 95.4 | 98.1 | 98.9 | 99.5 | 92.7 | 63.9 | 66.3 | 90.8 |
|  | Reproduced | 100.0 | 99.4 | 88.8 | 94.1 | 96.7 | 96.2 | 98.5 | 99.1 | 92.9 | **72.3** | *93.7* | 93.8 |

We highlight differences (w.r.t the original publication) greater 5% in **bold** and greater 10% additionally in *cursive*.

To investigate if the results are indeed connected to our implementation, we perform an additional experiment. We implemented a corrected version of the Blur + JPEG augmentation and rerun the experiment. The results are mixed. While we achieve a closer result for the DeepFake data set, the changes lead to even higher differences in the SAN and SITD data sets. We repeat the experiment two additional times to investigate if this might be related to chance. We obtained widely varying results on the SITD (84.8/85.9/90.7), the SAN (72.6/74.1/78.3), and the DeepFake (72.4/73.0/76.1) data sets. A full table can found in the supplementary material.

The authors already noted the SAN and DeepFake data set as exceptions. Our model recovers better from using a combination of blurring and JPEG compression for these sets. Thus, we hypothesize that our implementation may apply more aggressive preprocessing (we always apply blurring and JPEG compression), but on fewer training examples (our implementation is all or nothing; Wang et al. can apply blurring, JPEG compression, or both). The probabilistic nature of these data augmentations can also explain the fluctuation in the SITD, SAN, and DeepFake results. Obtaining statistical guarantees for our results would be ideal. However, we would need to perform multiple runs of ours' and Wang et al.'s implementation. With a single run already taking 42 GPU hours, we abstain from further investigation. We also do not investigate the hand-crafted blurring implementaion. We assume that future practitioners will preferably use the build-in PyTorch method we utilized (unavailable at the time of original publication). We hypothesize that our experiments better reflect future results.

### 3.2 Claim: *More diverse data improves generalization*

Second, we investigate how diversity of the data set affects the results by reproducing the rest of Table 2. Similar to the original paper, we train five different classifiers on subsets of the original training data ($\{2, 4, 8, 16, 20\}$-classes) and use blurring and JPEG compression with a probability of 50% during training. The results are displayed in Table 2.

**Results:** Overall, we can again recreate the general trend that data diversity improves generalization. The results mostly match, but we can again observe widely different results for the SITD, SAN and DeepFake data sets. We attribute these differences to the earlier observed sensitivity to the random character of these experiments.

The original publication observed that data diversity helped with generalization up to a specific point. The authors hypothesized that at this point (16 to 20 classes) the training set may be diverse enough for practical generalization. We agree that data set diversity helps with generalization. We cannot fully agree that there exists a point of diminishing returns. Utilizing more classes allows our classifiers to achieve significantly better results on the DeepFake ($76.8 \rightarrow 93.7$ AP) and SITD ($84.5 \rightarrow 92.9$ AP) data sets.

While the diminishing returns might be related to chance, we additionally investigate if they are connected to the aggressive data augmentations used in the original experiments. We perform a control experiment where we again train multiple classifiers on subsets of the training data with all augmentations disabled. Note the original publication only assessed subsets with augmentations enabled.

The results are presented in Figure 1, a full table can be found in the supplementary material. Contrary to our hypothesis, the data augmentation help the model generalize in the presence of more diverse data. However, they also lead to

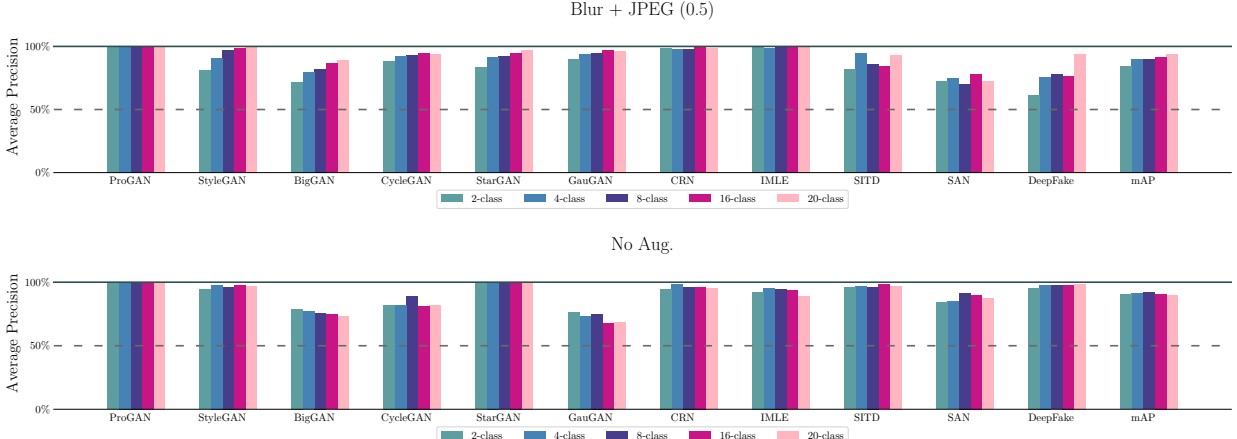

Figure 1: **Comparison of different data set sizes.** When training with augmentations we observe a steady increase in performance while increasing the diversity of the data set. We get mixed results for training without augmentations. Apparently, the higher data diversity can only be utilized with the aid of data augmentations. Best viewed in color.

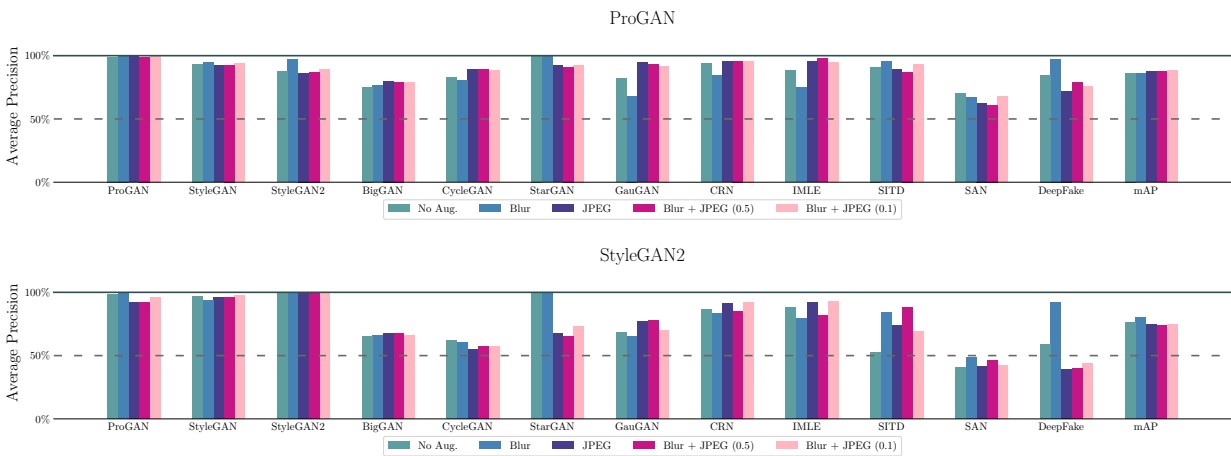

Figure 2: **Comparison between two training sets generated by ProGAN and StyleGAN2.** The classifier trained on data sets generated by ProGAN generalizes significantly better to other architectures. Blurring seems to help the classifier trained on StyleGAN2 to generalize better, the other augmentations hurt performance. Best viewed in color.

varying results on the SITD, SAN, and DeepFake data set. We assume the same probabilistic nature observed in Section 3.1 is related to the fluctuations. For the other data sets, we assume that the augmentations act as a regularizer, preventing the model from overfitting to the training set. We study this phenomena further in Section 3.3.1. Overall we conclude that more diverse data improves generalization.

## 3.3 Results beyond the original paper

We perform further experiments to get more insights into the original paper. First, we investigate if the results are dependent on the generator used to create the training set by sampling a second data sets from StyleGAN2 [Karras et al., 2020]. Second, we investigate if the results are dependent on the classifier used by comparing the ResNet50 against a different image classifier (VGG-11) and a classifier which leverages spectral analysis [Frank et al., 2020].

### 3.3.1 Research Question: *Do the results depend on the generator which creates the data set?*

We want to investigate if the generalization capabilities depend on the generator which creates the training set. As described in Section 2.1, we utilized pre-trained StyleGAN2 models to create a novel data set.

Table 3: **Comparison of different classifiers trained with different data augmentations.** We compare the AP and mAP across different classifiers which are trained on the 8-class data set. The Blur + JPEG augmentations refers to the 50% variant. Chance is 50%, the best possible results is 100%, and we highlight the ProGAN results in gray since the classifier is trained on similar data. For each classifier we highlight its best result per colum in **bold**.

| Classifier | Augmentation | ProGAN | StyleGAN | BigGAN | CycleGAN | StarGAN | GauGAN | CRN | IMLE | SITD | SAN | DeepFake | **mAP** |
|---|---|---|---|---|---|---|---|---|---|---|---|---|---|
| ResNet50 | No Aug. | 100.0 | **97.9** | 77.0 | 81.9 | **100.0** | 73.5 | 98.0 | 95.4 | 96.8 | **84.9** | 97.6 | **91.2** |
| | Blur | 100.0 | 97.1 | 71.6 | 84.7 | **100.0** | 67.0 | 78.2 | 76.4 | **97.6** | 75.5 | **98.0** | 86.0 |
| | JPEG | 100.0 | 96.7 | **83.5** | 91.6 | 90.2 | 95.9 | **98.8** | 99.4 | 86.1 | 71.6 | 85.2 | 90.8 |
| | Blur + JPEG | 100.0 | 96.7 | 81.8 | **92.7** | 92.6 | 95.0 | 98.2 | 99.3 | 85.7 | 69.8 | 77.9 | 90.0 |
| VGG-11 | No Aug. | 100.0 | 96.6 | 76.8 | 78.3 | **100.0** | 60.7 | 56.8 | 56.8 | 99.7 | **91.4** | **94.5** | 82.9 |
| | Blur | 100.0 | 94.9 | 79.1 | 81.2 | **100.0** | 56.6 | 51.6 | 51.5 | **99.9** | 82.2 | 92.1 | 80.9 |
| | JPEG | 100.0 | **98.7** | **87.3** | 91.1 | 97.6 | 85.8 | **98.8** | 99.1 | 94.1 | 80.3 | 91.1 | **93.1** |
| | Blur + JPEG | 99.8 | 94.5 | 85.3 | **93.3** | 89.3 | 91.2 | 98.4 | 98.0 | 95.0 | 68.2 | 77.5 | 90.1 |
| DCT-ResNet | No Aug. | 100.0 | 98.0 | **87.5** | 63.1 | 97.2 | 96.0 | 65.7 | 75.6 | 43.7 | 52.2 | 57.0 | 76.1 |
| | Blur | 100.0 | 97.5 | 82.5 | **71.7** | **99.5** | **96.9** | 71.4 | 86.0 | 49.2 | 47.4 | 59.3 | 78.3 |
| | JPEG | 100.0 | 93.1 | 69.0 | 50.8 | 58.0 | 43.9 | 79.8 | 95.7 | 96.5 | **61.4** | **71.9** | 74.5 |
| | Blur + JPEG | 100.0 | **98.2** | 86.8 | 54.4 | 88.5 | 74.9 | 82.0 | 95.7 | **97.9** | 59.3 | 68.7 | **82.4** |

We then trained five ResNet50 classifier using the augmentation proposed by Wang et al.. For comparison, we also trained five classifier on LSUN *cat and horse* data generated by ProGAN. In contrast to our previous experiments, we also include a StyleGAN2 test set which was provided by Wang et al. after the original publication. The results are displayed in Figure 2 and the corresponding Table can be found in the supplementary material.

**1) Data augmentations help ProGAN generalise to StyleGAN2:** When utilizing the proposed data set augmentation, the classifiers trained on ProGAN generalize to detect images from StyleGAN2 ($87.6 \to \{86.3, 87.1, 89.2, 97.1\}$ AP). However, the same does not apply for StyleGAN2. Data generated by StyleGAN2 already achieves quite a high performance on ProGAN, thus adding data augmentations does only slightly improve or even hurts performance.

**2) Data augmentations are specific to the data set:** Blurring seems to generally hurt the ProGAN classifier (Cycle-GAN, GauGAN, CRN, IMLE, SAN, and overall mAP), while it allows the StyleGAN2 classifier to generalize to the ProGAN, the StarGAN, the SITD, and, the DeepFake data set. It also achieves the best overall performance. Thus, data augmentations seem to be specific to the training data set used and should be evaluated carefully.

**3) ProGAN serves as a better prior for generalization:** If we compare the individual performance and the overall mAP, the classifiers trained on ProGAN seem to better generalize to other architectures. We hypothesize that this is due to the fact that StyleGAN2 is a very recent proposal (released in 2020), while all other generators were proposed earlier (2016-2019). ProGAN is itself an older generator (2018), thus the images generated by it might have more in common with older architectures.

We conclude that the results indeed depend on the generator used for creating the data set. The data augmentations do not transfer and our results imply that going forward, practitioners might have to continuously update their training data sets to generalize to new architectures. Note that we only performed experiments with two classes, adding more classes might improve the performance of StyleGAN2 as a training set.

### 3.3.2 Research Question: *Do data augmentations transfer to other classifier?*

Finally, we want to investigate the question if the results transfer to other classifiers. We train two other classifiers to compare our results: a VGG-11 model and a classifier which utilizes spectral information (DCT-ResNet). The results are depicted in Table 3, additionally, we provide a plot of the results in the supplementary material. When we report relative numbers, these are in reference to the results without augmentations (No Aug.).

When using (No. Aug/Blur) augmentations the VGG model is incapable of generalizing to the GauGAN (60.7/56.6 AP), CRN (56.8/51.5 AP) and IMLE (56.8/51.5 AP) data sets. This can be circumvented by utilizing the JPEG augmentation, which also leads to the overall best result (93.1 mAP). Yet, the augmentation does cause a drop in AP for the StarGAN, SITD, SAN and DeepFake data sets. This performance drop can also not be offset by using both augmentations in conjunction.

The performance of the DCT based classifier is worse overall (76.1/78.3/74.5/82.4 mAP). Blur seem to help general-ization, allowing for better results on CycleGAN, StarGAN, CRN and IMLE data sets. JPEG augmentation allows the classifier to drastically improve its results on SITD (+52.8 AP) and helps generalization to CRN (+14.1 AP), IMLE (+20.1 AP), SAN (+9.2 AP) and DeepFake (+14.9 AP). However, it severely hurts performance for Star-

GAN ($-39.2$ AP) and GauGAN ($-52.1$ AP), and, to a lesser extend, StyleGAN, BigGAN and CycleGAN. Combining both augmentations leads to a good compromise, achieving the best overall performance (82.4 mAP) and reducing the negative effects for StarGAN ($-8.7$ AP) and GauGAN ($-21.1$ AP).

Overall, we conclude that the proposed data augmentations transfer to other classifiers. However, there exists no silver bullet which helps generalization in general.

# 4  Discussion

Since we focused our efforts on reproducing the paper from scratch, we can only partially comment on the source code. However, we later used it to compare our implementation against. In the following, we want to give an overview of our observations.

## 4.1  What was easy?

The overall description in the paper is very detailed. The authors discuss in detail how they collected the data for the test set, provide a good description of their training procedure, and summarize their results clearly with the aid of tables and figures. Additionally, the authors include a long appendix with further details. We did find one description in the paper confusing, which we discuss in more detail in Section 4.2.

As discussed above, we only had a limited exposure to using the authors' code. The parts we looked at were of good code quality, however, sometimes could use more comments. The README file of the repository is very detailed, it includes clear instruction on how to set up the code, download the corresponding data, train models from scratch, and evaluate the results. Additionally, the authors include links to download pre-trained models. We used these models to validate our results and found them quite intuitive to use.

## 4.2  What was difficult?

We came across very few hurdles when reproducing the code. However, we did end up slightly miss-implementing one of the data augmentations. In the paper, the authors describe their introduced data augmentations in Section 4.1. The description for the *Blur+JPEG* augmentation states: "the image is possibly blurred and JPEG-ed, each with 50% probability". However, the shorthand explicitly uses a plus sign and in the remainder of the paper only the shorthand is used. Thus, based on the frequent use of the plus sign, we assumed the augmentation *always* applies the blurring and JPEG compression together. This resulted in us implementing a slightly different version of the data augmentation, where we apply both in conjunction with 50% probability. We want to stress that the paper states this correctly, we simply missed the last part of the description.

## 4.3  Communication with original authors

Due to the quality of the paper and the availability of the code repository, we could resolve every question. We did message the authors regarding our different results for the Blur and JPEG augmentation. They agreed that it is difficult to conclude an exact reason why and how augmentations help generalization. For further research direction, they suggested further research into which features the CNN learns.

# 5  Conclusion

In summary, we believe the paper to be reproducible. We successfully recreated the original experiments from scratch. While our results are different, the overall trend remains the same. However, our additional experiments revealed several limitations to these claims.

More diverse data sets help the classifier generalize to unseen architectures, but: the choice of generator is crucial and data set-specific augmentations have to be used. When using no augmentations, more data diversity hurts the performance of the classifier trained on ProGAN. When we instead use StyleGAN2 as the generator for the data set, we achieve significantly lower performance for all tested configurations. This suggest the possibility that future practitioners might have to constantly update their training sets with new architectures.

Also, the data augmentations seem to be specific for the training data set used. When using ProGAN generated data as the training set, the augmentations transfer across different classifiers. However, only blurring helps StyleGAN2 generalize, while the other proposed augmentations hurt performance. Thus, we advice practitioners to carefully investigate which augmentations help their current situation.

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
