# OpenReview forum: "[RE] CNN-generated images are surprisingly easy to spot... for now"
_ML_Reproducibility_Challenge/2020 — Reject_

### Official Review · AnonReviewer3 · 2021-02-23
**Very nice report.**

**Rating:** 9
**Confidence:** 3

**Review:**

The report aims to reproduce the results from the following paper "CNN-generated images are surprisingly easy to spot... for now". The report is very complete, well written and well executed and although I have not read the original paper I was able to understand the ideas behind the original paper just by reading the report.

**Familiar With The Original Paper:**

I have not read the original paper

**Reproducibility Summary:**

Report has summary

---

### Official Review · AnonReviewer2 · 2021-02-28
**A comprehensive reproductibility report**

**Rating:** 8
**Confidence:** 2

**Review:**

This report reimplements the algorithm proposed in the original paper and confirms the validity of the experimental results in the paper. The pre-trained model used in this reproducibility assessment paper can be accessed in an anonymous git repo and the report gave details about how the reproducibility test is organized. Additionally, the report also points out the performances in the original paper depend on the choice of the training data set and the choice of the data generator.  This is the limit of the work in the original paper, which helps practitioners to better use the proposed algorithm.



**Familiar With The Original Paper:**

I have read the original paper

**Reproducibility Summary:**

Report has summary

---

### Official Review · AnonReviewer1 · 2021-03-06
**Borderline submission; could go either way**

**Rating:** 5
**Confidence:** 5

**Review:**

The paper seeks to reproduce the results of the paper titled ‘CNN-generated images are surprisingly easy to spot... for now’. They sought to validate two  main claims in the original paper – that data augmentation and data diversity helps with generalization in the context of real/fake image classification. They don’t validate the third claim that data augmentation aids robustness. In addition to the above, the paper also investigates the situations then the training data generator and classifiers are changed.

Overall, the reproducibility study is reasonable. And, despite the shortcomings mentioned below, it should be useful to the audience of this challenge as well as the wider AI/ ML community interested in the original work. However, due to the concerns below, I recommend rejecting this paper with a rating of marginally below threshold (which I’m willing to revise based on discussions with the authors).

In the following, an evaluation of this paper on the metrics suggested by the RC 2020 challenge is presented:
Reproducibility Summary:

The authors have provided a brief and clear summarization of the problem statement and the proposed approach and have reported their major findings.

Scope of Reproducibility:

The authors clearly enumerate the scope of the reproducibility study to validate two claims from the paper with a partial recreation of the original experiments as well as several additional ones. The study is designed and conducted accordingly.
Code:

The code for the original paper is publicly available. However, the authors have re-implemented the code for the experiments performed and use the original implementation sparingly. The code base is submitted with readable code and docs.
Communication with original authors

The authors mention that most experiments could be reproduced using minimal communication with the original authors, given the details in the main paper and the well-documented code repository. There were a few experimental settings that lacked clarity (or were misunderstood by the report authors) which were resolved by communicating with the original authors.

Hyperparameter search

For the reproducibility experiments, the authors use the hyperparameter details provided in the main paper and do not perform any parameter tuning. However, the authors perform data augmentation (Blur + JPEG) differently from that of the main paper. The variation in the implementation is also hypothesized as the reason for the discrepancy between the obtained and original results. Thus, a hyperparameter tuning for the same is also expected.

A discussion on the hyperparameter search for the additional experiments performed is missing.
Discussion of results

I have the few concerns:

Data diversity and generalization: These results are presented in Table 1. The authors conclude (line 154) that these results are similar to the original reported results. This is clearly not the case when differences as high as 24% in absolute terrms (Deepfake 20-class: 66.3% --> 93.7%; SAN 2-class: 52.9 --> 72.3) are seen. Even when the differences are low, they seem statistically significant (StarGAN 2-class: 87.3 --> 83.5). However, the trend that diversity leads to better generalization seems to hold.

It is speculated that the difference in results is due to the different blurring function used and the way blurring, and JPEG compression is applied to the data. This could be the reason but doesn’t seem to be verified.

The authors also demonstrate via Figure 1 that the correlation of improved performance with more diversity is stronger when accompanied by data augmentation (blur, jpeg compression) than without.

The observation “dependent on the data set” in lines 173-174 is concerning. Since this reference to the data set is test data, all it says is that the results are not expected to hold across different test scenarios.

Data augmentation and generalization: This analysis is reproduced in Table 2. We observe statistically significant differences even in this experiment, even for the ‘No Aug’ scenario where no blur or JPEG compression is applied. For example, (SAN No Aug: 93.6 --> 87.2). Secondly, the trend of improved performance with data augmentation is really mixed with 4 settings out of 10 -- StarGAN, SITD, SAN and DeepFake bucking the trend. Whether this can be entirely attributed to variations in implementation is not investigated

However, the overall trend that diversity (when accompanied by augmentations) shows an improvement in performance seems clear and stands validated by the report.

The authors also make a key observation that data diversity alone (without augmentation) is insufficient to achieve generalization which was not clear from the experiments in the original paper.
Recommendations for reproducibility
The authors were able to reproduce two major claims of the paper and obtain better performance than originally reported for some experiments. They suggest that performing the data augmentation – “blur + JPEG,” simultaneously rather than sequentially helps improve overall performance.
Results beyond the original paper:
Two additional investigations are carried out that go beyond the original paper: (a) changing the generator used for training (pre-trained StyleGAN2), and, (b) training a different classifier (VGG and DCT-ResNet).

While some trends are clear (ProGAN better than StyleGAN2 for training), others not quite and it seems that the authors are not careful in making deductions.
-	Line 206: it should be 87.6 --> (86.3, 87.1, 89.2, 97.1}
-	(l. 209-212): For StyleGAN2, other proposed augmentations don’t worsen results for StyleGAN2: not for GauGAN, CRN, IMLE, SITG, SAN. For ProGAN: blurring doesn't hurt for StyleGAN, StyleGAN2, BigGAN, StarGAN, SITD, DeepFake. Other's don't improve for StarGAN, SAN, DeepFake.

The question of whether the performance improvement and generalization trends hold for other classifiers is not properly discussed. It also lacks a detailed discussion on the implementation and hyperparameter tuning for the additional experiments performed.
Overall clarity and organization

The report is well structured in general and has a reasonable clarity. The readability could have been improved by organizing the experiments in the report in the same order as that of the main paper (where discussion on data augmentation precedes diversity).

The authors should address the following issues in their draft:
-	Typos in specifying the learning rates (lines 124-126): 1e-3 or 10^-3 instead of 1^-3 etc.
-	(line 10) It’s not clear what is meant by - “if the results extend beyond the original contribution”.
-	The authors don’t clearly mention whether the data augmentations were applied only to the synthetic images or the real images as well.


**Familiar With The Original Paper:**

I have read the original paper

**Reproducibility Summary:**

Report has summary

---

### Decision · Program_Chairs · 2021-03-31

**Decision:**

Reject

**Comment:**

Overall reviews and/or the paper content not good enough for the AC to recommend to the journal.